# Study on Axial Tensile Strain Rate Effect on Concrete Based on Experimental Investigation and Numerical Simulation

**DOI:** 10.3390/ma15155164

**Published:** 2022-07-25

**Authors:** Bi Sun, Rui Chen, Yang Ping, Zhende Zhu, Nan Wu

**Affiliations:** 1Harbin Institute of Technology (Shenzhen), Shenzhen 518055, China; sunbi58@126.com (B.S.); cechenrui@hit.edu.cn (R.C.); 2Shenzhen Water Planning and Design Institute Co., Ltd., Shenzhen 518001, China; 3PowerChina Eco-Environment Group Co., Ltd., Shenzhen 518102, China; pingy@swpdi.com; 4Key Laboratory of Ministry of Education of Geomechanics and Embankment Engineering, Hohai University, Nanjing 210098, China; zhendezhunj@163.com; 5Guangzhou University-Tamkang University Joint Research Centre for Engineering Structure Disaster Prevention and Control, Guangzhou University, Guangzhou 510006, China

**Keywords:** concrete, interfacial transition zone (ITZ), tensile strain rate, meso-strength parameters, particle flow code, intermediate strain rate

## Abstract

The material of concrete is a three-phase composite material composed of an aggregate, a mortar and an interface transition zone (ITZ). Based on this characteristic, the axial tensile test of mortar, the interface and concrete specimens under intermediate strain rate was carried out in this paper. The sensitivity of these three materials to strain rate was compared and analyzed. The numerical simulation of the axial tension of the concrete materials was studied. The following conclusions are drawn: in the axial tension test, the rate of sensitivity of the specimen interface is the strongest. With the increase in strain rate, the tensile strength and elastic modulus of concrete specimens increase but the effect of the ITZ decreases. The low tensile strength of the ITZ leads to its failure in concrete. The parallel bond strain energy and the dissipated energy of specimens increase with the strain rate. When the strain rate is higher (greater than 1 × 10^−2^), the increase rate of the dissipated energy is greater than that of the parallel bond strain energy. The results of this research can provide the corresponding basis for the safety evaluation and the stability analysis of concrete engineering in the range of intermediate strain rate.

## 1. Introduction

Concrete is widely used in geotechnical engineering, such as dams, tunnels, high-rise buildings and nuclear power stations. The experimental and numerical simulation results of the meso-structure of concrete show that [1,2,3,4,5,6,7,8] the interface transition zone (ITZ) between the aggregate and the cement matrix of concrete is very different in physical and mechanical properties.

The mechanical properties of ITZ depend on many aspects under the condition of static loading [9,10,11,12,13,14,15,16,17,18,19]. Zhang et al. [20] thought that the outer modified layers endow a dense microstructure and high bond strength of the ITZ between high-performance lightweight aggregates and cement paste. Jebli et al. [21] indicated that the tensile strength of the cement–aggregate interface was about 30% lower than that of the cement paste tensile strength. Xi et al. [22] studied the meso fracture characteristics of the ITZ with an artificial neural network and thought that the ITZ significantly affected the cracking behavior of concrete. Wu et al. [23] believed that the improvement of the ITZ properties can contribute the enhanced behavior of the recycled aggregate concrete. Song et al. [24] held that the ITZ is often considered to be the weakest region in concrete. Therefore, it is generally agreed that the ITZ governs concrete strength [25,26].

The discrete element method (DEM) has been widely used to study the mechanical behavior of soil, rock and concrete because it can visually express the nonlinear damage process of materials with element contact [27,28]. Zhou et al. [29,30] studied the influence of the ITZ strength on the failure mode, number of microcracks, tensile strength and strain rate effect of concrete by using DEM. Zhao et al. [31] showed that the elastic modulus of concrete is insensitive to the strength of the ITZ. In 1979, Cundall et al. [32] proposed the particle flow code (PFC) using a particle disk based on the DEM in order to study the mechanical behavior of granular media. Zhang et al. [33] considered the ITZ to directly affect concrete strength, stiffness and durability by using PFC. Now, PFC has become an important tool for the study of granular materials. It has been widely used in mining, geotechnical engineering and many other fields [34,35,36,37,38,39,40].

Generally, concrete materials are not sensitive to low strain rates (strain rates less than 10^−5^) but very sensitive to high strain rates (strain rates greater than 10^2^), while intermediate strain rates are located in the transition zone where brittle materials are sensitive to strain rates [41,42,43]. Therefore, it is necessary to study the mechanical response characteristics of the concrete materials under intermediate strain rate. Due to the limitations of test conditions and technical levels, there are few studies on the axial tensile mechanical properties of the concrete materials under intermediate strain rate. Further exploration and deepened research is needed regarding many aspects [44]. The concrete material is a three-phase composite material composed of aggregate, mortar and ITZ. In view of this characteristic, the axial tensile tests of mortar, interface and concrete specimens were carried out under medium strain rate in this paper. The sensitivity of the three materials to strain rate was compared and analyzed. A meso model of the concrete was established by PFC2D. The evolution of concrete microcracks and energy under different strain rates was studied. The research results can provide the corresponding basis for the safety evaluation and stability analysis of concrete engineering within the range of medium strain rate caused by mining, vehicle driving, landslide, seismic disturbance, etc.

## 2. Axial Tension Test

### 2.1. Specimens

The axial tension tests of mortar, interface and concrete specimens, and the uniaxial compression tests of mortar specimens were conducted. The concrete used in the test was mixed at a proportion of 0.52:1:1.67:2.47 (water:cement:sand:aggregate). The mortar parts of the mortar and interface specimens were made of a slurry of concrete. The concrete and mortar were poured in a steel mold with a size of 1100 mm × 300 mm × 300 mm and were cured under standard conditions for 28 days. The concrete and mortar materials were drilled into cylinder specimens with a diameter of 74 mm. Then the drilled specimens were cut to a height of 150 mm. The concrete specimen is shown in Figure 1a. The interface specimens consist of rhyolite and mortar. One end of the rhyolite, with a diameter of 74 mm, was cut flat and the other end was a natural section, as shown in Figure 1b. The rhyolite core was tightly wrapped into a cylinder by a rigid plastic plate. The core was placed in one end of the cylinder and its natural section faced inward. The cylinder was vibrated by the shaking table after pouring the mortar into the cylinder. The cylinder specimens were cut and polished at a size of 74 mm × 150 mm, as shown in Figure 1c. The steel plates were glued to both ends of the specimen to fix it to the test equipment. The compression test of the mortar was a supplementary test. The height of each specimen was 100 mm and the diameter was 50 mm. There were nine mortar, interface and concrete specimens used for axial tension test, respectively, a total of 27. There were three specimens used for the compression test.

### 2.2. Test Equipment

The tests included a uniaxial compression test and a uniaxial tension test. The tests were carried out on a servo controlled electrohydraulic material testing system (MTS 322) equipped with two manufactured spherical joints to reduce eccentricity in testing, as shown in Figure 2a. The load transferred to the specimen by the spherical joints and screws. Across the lateral, three LVDTs (linear variable differential transducer) were attached to the concrete with a measuring length of 140 mm under 120° between each other. The three strain values were recorded. Additionally, the strain values were averaged and fed into a variable-gain amplifier, so a specific strain rate was achieved. The uniaxial compression test was carried out on RMT-150B multi-functional full-automatic rigid rock servo testing machine, as shown in Figure 2b.

### 2.3. Effect of Strain Rate on Tensile Strength

The tensile strength is linearly related to the strain rate when the strain rate range is 10^−6^–10^−1^ s^−1^ [43]. The different loading rates result in different strain rates. In this tensile test, three specimens were subjected to tensile test under the same loading rate level. There were three loading rate levels for the test. Therefore, nine specimens of each material were subjected to the test. Therefore, there were nine specimens of each material available for testing. The loading rates of concrete specimens were 10^−3^ m/s, 10^−4^ m/s and 10^−5^ m/s, respectively. The loading rates of mortar and interface specimens were the same, 10^−4^ m/s, 10^−5^ m/s and 10^−6^ m/s, respectively. The tensile test results of mortar, interface and concrete specimens under different strain rates are shown in Table 1.

The test results show that the peak stress of the three kinds of specimens increases with the strain rate, and the strain at the peak stress of mortar and interface tends to increase with the strain rate as well. The elastic modulus of all specimens changes slightly with different strain rates. Under the same loading rate, the mortar and interface specimens have the largest and smallest peak stress, respectively. To simplify the calculation, it is approximated that the relationship between the ratio of the tensile strength and the logarithm of the strain rate ratio is linear as Equation (1).
(1)$$\frac{f_{tq}}{f_{ts}}=1+\alpha \log\left(\frac{\dot{\varepsilon }}{{\dot{\varepsilon }}_{s}}\right)\;\;\;\;\;\;\;\left(\dot{\varepsilon }<1\times 10^{-3}\right)$$
where $f_{tq},f_{ts}$ are quasi-static and the static tensile strength of the specimens, respectively. $\dot{\varepsilon }$ is quasi-static strain rate, and ${\dot{\varepsilon }}_{s}$ is static strain rate. The static strain rates of mortar, interface and concrete specimens are approximately 8 × 10^−^^7^, 6 × 10^−^^7^, 7 × 10^−^^6^, respectively. α represents the strain rate sensitivity coefficient of the tensile strength. According to the fitting results of test data, the α of mortar, interface and concrete specimens are 0.2368, 0.5375 and 0.1991, respectively.

According to the fitting results, the *α* of the interface specimen is the largest, that is, the rate sensitivity of interface specimen is the strongest. Generally [41,42], the larger strain rate would result in more broken aggregates. In other words, the influence of aggregate on the strength of concrete increases with the strain rate, while the influence of ITZ decreases. On the macro scale, the rate sensitivity of concrete is determined by the combined action of mortar, ITZ and aggregate. The influence degree of each component of concrete on its rate sensitivity also varies with the change of strain rate. In the test results of this paper, the strain rate sensitivity coefficient of the mortar specimen is stronger than that of the concrete.

The results show that the elastic modulus tends to increase with the strain rate, and the relation can be expressed as Equation (2):(2)$$E_{tq}=b+\beta \log\left(\frac{\dot{\varepsilon }}{{\dot{\varepsilon }}_{s}}\right)\;\;\;\;\;\;\;\left(\dot{\varepsilon }<1\times 10^{-3}\right)$$
where $E_{tq}$ is the quasi-static elastic modulus under direct tension, β represents strain rate sensitivity coefficient of the quasi-static elastic modulus, and b is constant. According to the formula of Equation (2), the β and b values fitted by the results of M4-3, M5-3 and M6-3 mortar specimens are 0.8656 and 21.886, and the β and b values fitted by the results of C3-3, C4-1 and C5-3 concrete specimens are 1.1455, 30.295.

### 2.4. Effect of Strain Rate on the Stress-Strain Curves

Figure 3 shows three typical stress–strain curves of specimens under tension with different strain rates. For the selected typical curve, two aspects were considered: whether the peak stress is closer to the average value and whether the contrast is obvious.

From Figure 3, the tensile strength increases basically with the strain rate for the same type of specimens. The elastic modulus changes slightly. 

## 3. Numerical Modeling and Parameter Calibration

### 3.1. Contact Model

In this research, the linear parallel bond model (PBM) was chosen for the numerical simulation, because it is more realistic for rock-like material modeling, in which the bonds may break in either tension or shearing with an associated reduction in stiffness [45,46]. The relation can be expressed as Equation (3) [47]:(3)$$F_{c}=F_{l}+F_{d}+\bar{F},\;{\bar{M}}_{c}=\bar{M}$$
where $F_{l}$ is the linear force, $F_{d}$ is the dashpot force, $\bar{F}$ is the parallel bond force and $\bar{M}$ is the parallel bond moment.

The PBM provides the behavior of two interfaces: The first interface is equivalent to the linear model. The linear forces of linear models are generated by linear springs with constant normal and shear sensitivities (normal stiffness $k_{n}$, shear stiffness $k_{s}$). The linear springs cannot sustain tension, and the slip is accommodated by imposing a Coulomb limit on the shear force using the friction coefficient μ. The second interface is called a parallel bond. The parallel bonding force of the PBM is produced by the combined action of the parallel bond stiffness (normal stiffness ${\bar{k}}_{n}$, shear stiffness ${\bar{k}}_{s}$), the parallel bond tensile strength ${\bar{\sigma }}_{c}$, the parallel bond cohesion $\bar{c}$ and the parallel bond friction angle $\bar{\phi }$. When the second interface is unbonded, the linear parallel bond model is equivalent to the linear model. 

### 3.2. Parameter Calibration of Mortar

Research [48,49,50,51] shows that micro-parameters determine the macro response of the granular material. The discrete element simulation can be implemented successfully only when the micro-parameters are selected correctly. The relationship between macro- and meso-parameters can be obtained by using the control variable method. Firstly, the meso-parameters of mortar were studied. To obtain the material property parameters for numerical calculation, uniaxial compression tests of mortar specimens were performed. The method of displacement control was adopted for loading. Three mortar specimens were tested, the loading rate was 10^−3^ m/s. The average compressive strength of the three specimens was 74.22 MPa. The data of a specimen with compressive strength close to the average value was adopted. The elastic modulus, compressive strength and Poisson’s ratio of the specimens were measured as 21.32 GPa, 73.17 MPa and 0.216, respectively.

1. The direct tension was calculated first to calibrate the tension modulus $E_{t}$ by changing the parallel bond effective modulus ${\bar{E}}_{p}$, as shown in Figure 4.

2. To study the effect of the linear bond effective modulus, $E_{l}$, on the compression modulus $E_{c}$, the value of ${\bar{E}}_{p}$ should be kept unchanged, as shown in Figure 5.

3. The value of ${\bar{E}}_{p}$ and $E_{l}$ should be kept unchanged to study the relationship between Poisson’s ratio υ and parallel bond stiffness ratio ${\bar{k}}_{n}/{\bar{k}}_{s}$ by biaxial compression. Since the linear model only works after the parallel bond breaks, the stiffness ratio of the linear model and the PBM are the same, as shown in Figure 6.

4. The ratio of the parallel bond tensile strength ${\bar{\sigma }}_{c}$ and the parallel bond cohesion $\bar{c}$ is defined as the bond ratio ${\bar{r}}_{s}$. When the shear bond between particles is broken, shear cracks appear. When the tensile bond is broken, tensile cracks occur. Therefore, when ${\bar{\sigma }}_{c}$ is too small, tensile cracks appear easily. Similarly, when $\bar{c}$ is too small, shear cracks appear easily. The propagations of different crack types lead to different failure modes of specimens. After determining the value of ${\bar{E}}_{p}$, $E_{l}$ and υ, the failure modes of specimens with different values of ${\bar{r}}_{s}$ are studied by biaxial compression, as shown in Figure 7.

In Figure 7, the red lines are shear cracks and the black lines are tensile cracks. In Figure 7b, there are only tensile cracks and no shear cracks. The macro-cracks formed by the meso-cracks are mainly distributed along the longitudinal direction. The tensile crack of the specimen is caused by splitting failure. In Figure 7c–f, shear cracks occur on the specimens. The shear cracks preferentially propagate along the inclined direction of the specimens, and tensile cracks also occur in this direction. At the same time, tensile cracks caused by splitting failure also appear in the longitudinal direction. In Figure 7g,h, the specimen displays shear failure, so the macro-cracks formed by the micro-cracks develop along the inclined direction. Shear failure occurs mainly in the specimen, so the macro-cracks formed by the meso-cracks develop along the inclined direction. According to the failure model of mortar in laboratory conditions, ${\bar{r}}_{s}$ is 1.2 in this paper.

5. According to the condition that ${\bar{r}}_{s}=1.2$, suppose that $\bar{c}=1\;\mathrm{MPa}$, then ${\bar{\sigma }}_{c}=1.2\;\mathrm{MPa}$, which is set as the reference bond strength. Multiplied by the coefficients of 0.5, 1.0, 2.0, 5.0, 10.0, 20, 30, 40 and 50 on the basis of the reference bond strength, different peak strengths will be obtained, respectively, as shown in Figure 8.

To sum up, the meso-parameters of mortar can be obtained: ${\bar{E}}_{p}=32.43\;\mathrm{GPa}$, $E_{p}=30.02\;\mathrm{GPa}$, ${\bar{k}}_{n}/{\bar{k}}_{s}=1.54$, ${\bar{r}}_{s}=1.2$, $\bar{c}=6.88\;\mathrm{MPa}$ and ${\bar{\sigma }}_{c}=8.26\;\mathrm{MPa}$.

### 3.3. Parameter Calibration of Concrete

According to the test results in Section 2.3, the tensile strength of different specimens at the same strain rate can be obtained by the conversion of Equation (1). When the strain rate is 6 × 10^−5^, the quasi-static tensile strength of mortar, the interface and the concrete specimens are 3.16 MPa, 2.10 MPa and 2.31 MPa, respectively. $\bar{c}$ and ${\bar{\sigma }}_{c}$ of the mortar are 6.82 MPa and 8.18 MPa, respectively, and those of the interface specimens are 4.53 MPa and 5.43 MPa, respectively. The ${\bar{E}}_{p}$ and $E_{l}$ of the ITZ are 0.7 times the mortar through repeated trial and error. The meso parameters of the aggregate can be obtained according to Qin’s calibration method [52]: ${\bar{E}}_{p}=63.94\;\mathrm{GPa}$, $E_{l}=218.86\;\mathrm{GPa}$, $k_{n}/k_{s}=1.15$, $\bar{c}=19.08\;\mathrm{MPa}$ and ${\bar{\sigma }}_{c}=22.90\;\mathrm{MPa}$.

The numerical model of the concrete was built using a CT scanning image of the specimen section, as shown in Figure 9a,b. The axial tension numerical simulation shows that the cracks gradually develop from the ITZ as shown in Figure 9c. Figure 9d shows the fracture section of the concrete after tensile failure, in which the area encircled by red lines is the cementation surface between aggregate and mortar (namely, the ITZ).

## 4. Meso-Mechanical Analysis of Concrete under Different Strain Rates

The tensile strength and elastic modulus of mortar and interface specimens under different strain rates can be obtained by fitting according to Equations (1) and (2). The corresponding meso-parameters are obtained from Section 3.2, as shown in Table 2.

### 4.1. Stress–Strain Curve

After calibrating the meso-parameters of the materials under different strain rates, the stress–strain curves of concretes can be obtained by numerical simulation, as shown in Figure 10. The numerical simulation of axial tension was carried out with strain rate in the range of 10^−5^–10^−1^ s^−1^. With the increase in strain rate, the tensile strength and elastic modulus of concretes also increase. The numerical results are in good agreement with the experimental results. The stress–strain curve fluctuates greatly when the strain rate is 10^−1^, which is mainly due to the violent shock of particles and the rapid failure of concrete under high strain rate.

The tensile strength and elastic modulus of concrete under different strain rates can be obtained by fitting according to Equations (1) and (2), and the corresponding parameters can be obtained by numerical calculation. The relative errors between the fitting and numerical result are shown in Table 3.

In Table 3, the fitting values under different strain rates are the expected values of the numerical value. The margin of deviation is 10% of the expected value. Only when the strain rate is 1 × 10^−1^ in the table, is the root mean square error (RMSE) greater than the margin of deviation. This is because the increate effect of specimen strength is more obvious under a higher strain rate. When the strain rate is in the range of 1 × 10^−5^~10^−2^, the RMSE is within the range of the deviation margin.

When the maximum parallel bond effective modulus of the ITZ is not greater than that of mortar, the larger the parallel bond effective modulus ${\bar{E}}_{p}$ of the ITZ would result in the smaller tensile strength of concrete until the fitting of the interface specimen is reduced and vice versa. Combining with the data in Table 3, it can be seen that with the increase of strain rate, the role of the ITZ decreases gradually. However, the ${\bar{E}}_{p}$ of the ITZ has not been adjusted by the change of strain rate during actual numerical calculation, which results in lower or higher strain rates and increasing errors. The numerical calculation of the elastic modulus is in good agreement with the fitting value. This is because the properties of mortar have the greatest influence on the elastic modulus of concrete, whether in the laboratory tensile test or numerical simulation.

### 4.2. Development of Micro-Crack

In order to analyze the evolution of crack initiation and the propagation of concrete specimens, the cracks are traced in the simulation of axial tension with strain rate of 1 × 10^−1^, as shown in Figure 11.

As shown in Figure 11a, when the stress reaches 4.15 MPa (95% peak stress), tensile failure occurs firstly at the interface between aggregate and mortar at the edge of the specimen. With the increase of tensile stress, the crack continues to propagate transversely to the nearest ITZ. The crack propagation when the stress reaches 4.35 MPa (100% peak stress) is shown in Figure 11b. When the stress drops to 3.04 MPa (70% peak stress after peak), the first crack band continues to propagate transversely to the nearest ITZ, and another new crack band near the ITZ appears, which tends to propagate towards the first crack zone, as shown in Figure 11c. When the stress decreases to 0.41 MPa (10% peak stress after peak), the two crack bands are connected. The second crack band continues to propagate until the specimen fails completely. Combined with the analysis in Figure 11 and the above analysis, it can be seen that in the range of intermediate strain rate the tensile failure of concrete is due to the low tensile strength of ITZ.

### 4.3. Evolution of Energy

In the numerical simulation, the parallel bond strain energy and dissipated energy are tracked to explore the energy evolution law and obtain a further understanding of the crack propagation process. The dissipated energy includes kinetic energy, sliding energy and damping energy.

As shown in Figure 12a, the parallel bond strain energy increases to a maximum of 0.442 N·m at the stress of 4.15 MPa (95% peak stress), while the dissipated energy is very small. The curve of the strain energy is gentle at the initial stage of loading and increases rapidly whereafter. This is because the specimen requires less energy at the initial stage of loading and the bond between particles must be broken before deformation at a later stage. The strain energy decreases, and the dissipation energy increases with continuing tensile loading because of the generation of micro-cracks. The parallel bonding strain energy is the work done to overcome the bond between particles. The micro-cracks develop further under the strain energy after the initiation. Therefore, the strain energy stored in the specimen is dissipated by damage.

Compared with Figure 12a,b, it can be seen that the strain energy decreases before the stress reaches the peak at the strain rate 1 × 10^−1^, while the strain energy decreases when the stress approaches the peak at the strain rate 1 × 10^−2^. This is because the failure of specimen is caused by the generation and propagation of micro-cracks. It takes time for a micro-crack to propagate, but there is not enough time for it at a strain rate 1 × 10^−1^. The stress at strain rate 1 × 10^−2^ will continue to increase, but the strain energy stored in the specimen will release and transform into dissipated energy, resulting in more micro-cracks. This is also the reason why the higher the strain rate, the greater the fragmentation degree of the specimen.

Figure 12c shows that the parallel bond strain energy and dissipated energy increase with the strain rate. However, when the strain rate is greater than 1 × 10^−2^, the increase rate of dissipated energy is greater than that of the parallel bond strain energy. This is because the peak values of parallel bond strain energy and stress are almost at the same time at a lower strain rate (less than 1 × 10^−2^). Therefore, the specimen has enough time for crack propagation and failure, and the energy aggregation and dissipation can maintain a uniform speed. When the strain rate is higher (greater than 1 × 10^−2^), there is not enough time for crack propagation. The continuous increase of stress leads to more micro-cracks in the specimen. The more micro-cracks, the more energy dissipated.

## 5. Discussion

The elastic modulus of the ITZ affects the tensile strength of concrete in Section 4.1. The relationship between the parallel bond elastic modulus ratio of the ITZ to the mortar and the tensile strength of concrete is shown in Figure 13. It is found that the smaller the elastic modulus of the ITZ is, the softer the interface transition zone is and the earlier the mortar cracks initiate along the same crack path with the ITZ, which means a larger influence of mortar would result in a larger tensile strength of concrete. The larger the elastic modulus of the ITZ, the harder the ITZ will be and the earlier the ITZ will crack. Early cracking in the ITZ reduces the tensile strength of concrete. Therefore, the elastic modulus of the ITZ can be used to analyze the influence of the ITZ and mortar on concrete when studying the influence degree of the component material.

With the increase of strain rate, the mortar begins to break before the ITZ breaks. This means the ITZ is becoming softer and softer, and the tensile strength of concrete is increased on the macro scale. When the strain rate increases to a certain extent, even the aggregate begins to break before the mortar break, and the tensile strength of concrete will further increase. This is consistent with the analysis in Section 2.1.

## 6. Conclusions

Based on the characteristics of the main components of concrete, axial tension tests of the mortar, interface and concrete specimens under intermediate strain rate were carried out in this paper. The meso-parameters were calibrated by the macro-parameters obtained from the experiments. The numerical calculation under different strain rates was studied. The following conclusions are as follows:

According to the direct tension test, the strain rate sensitivity of the interface specimen is the strongest, and that of concrete specimen is the weakest. With the increase of strain rate, the influence of ITZ on concrete strength decreases, while the influence of mortar on concrete strength increases. At the same time, the peak stress and the elastic modulus of the specimens also increase.With the increase of strain rate, the tensile strength and elastic modulus of concrete specimens become larger and larger, and the role of ITZ becomes smaller and smaller when the strain rate is in the range of 10^−5^–10^−1^ s^−1^.The low tensile strength of the ITZ results in the initial failure of the ITZ in concrete, which reduces the tensile strength of concrete. The parallel bond strain energy and dissipated energy of specimens increase with the strain rate. When the strain rate is higher (greater than 1 × 10^−2^), the increase rate of dissipated energy is greater than that of the parallel bond strain energy.

## Figures and Tables

**Figure 1 materials-15-05164-f001:**
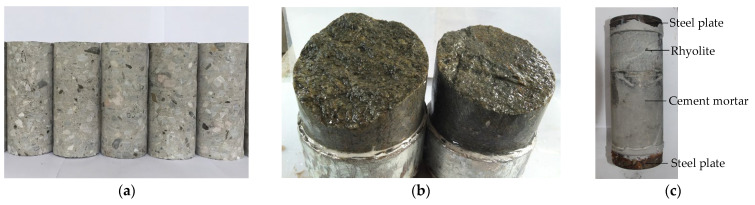
Interface specimens. (**a**) concrete specimen; (**b**) natural section of rhyolite; (**c**) interface specimen.

**Figure 2 materials-15-05164-f002:**
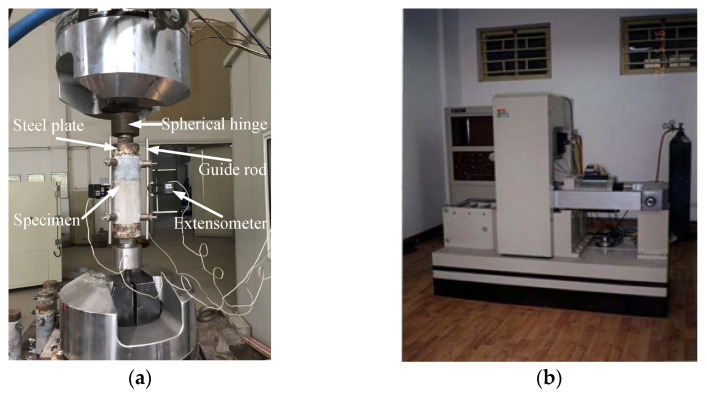
Loading equipment: (**a**) MTS 322; (**b**) RMT-150B.

**Figure 3 materials-15-05164-f003:**
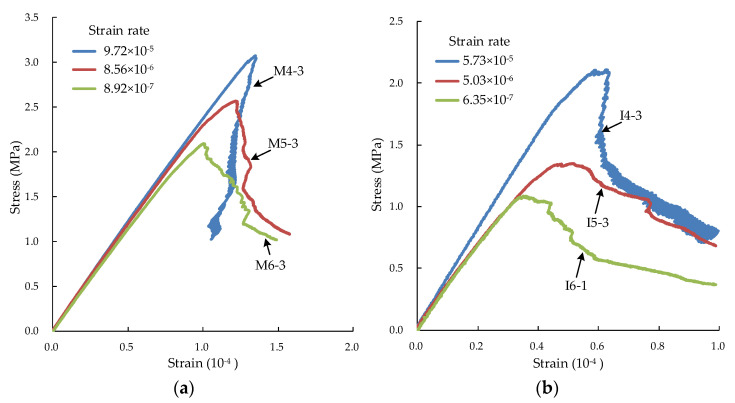
Stress–strain curves of specimens under different strain rates. (**a**) Mortar; (**b**) interface; (**c**) concrete.

**Figure 4 materials-15-05164-f004:**
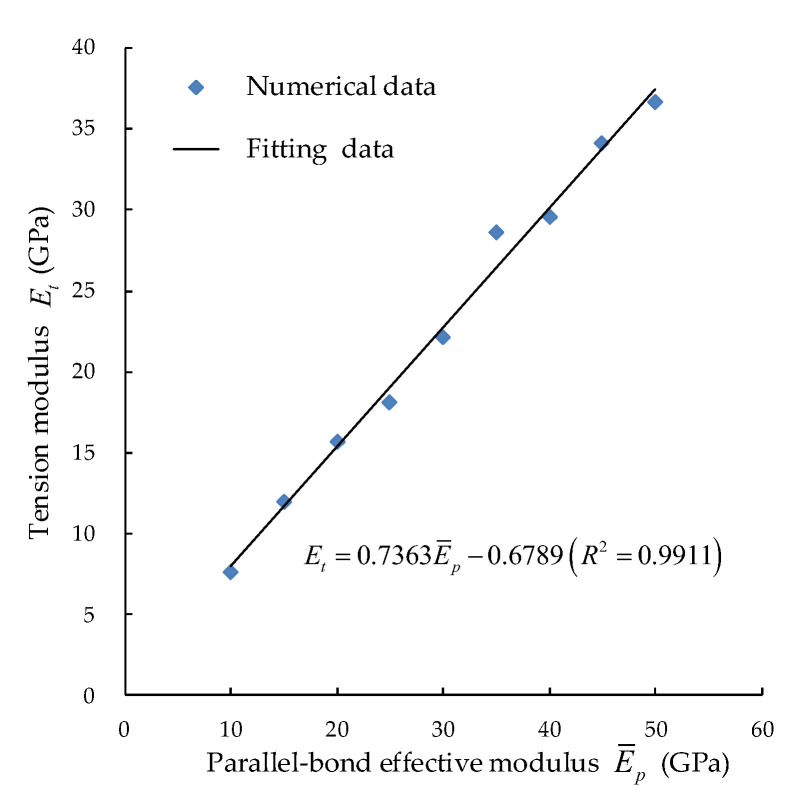
Corresponding relation between tensile elastic modulus and parallel bond effective modulus.

**Figure 5 materials-15-05164-f005:**
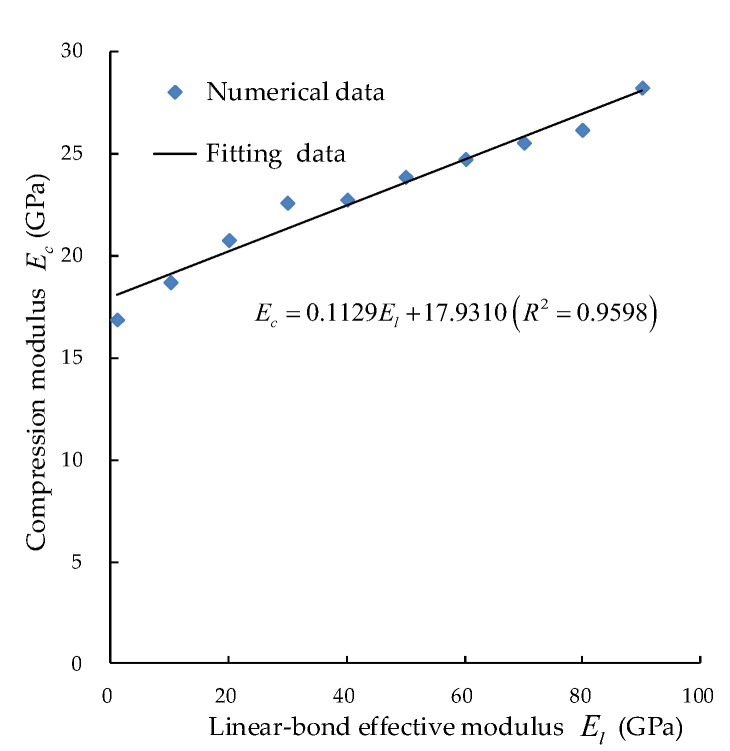
Corresponding relationship between the compression modulus and the linear bond effective modulus.

**Figure 6 materials-15-05164-f006:**
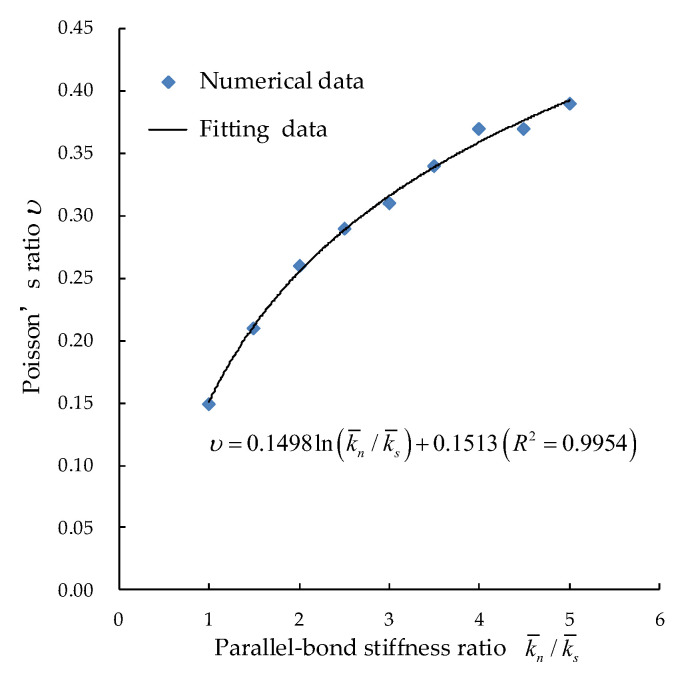
Relation between Poisson’s ratio and parallel bond stiffness ratio.

**Figure 7 materials-15-05164-f007:**
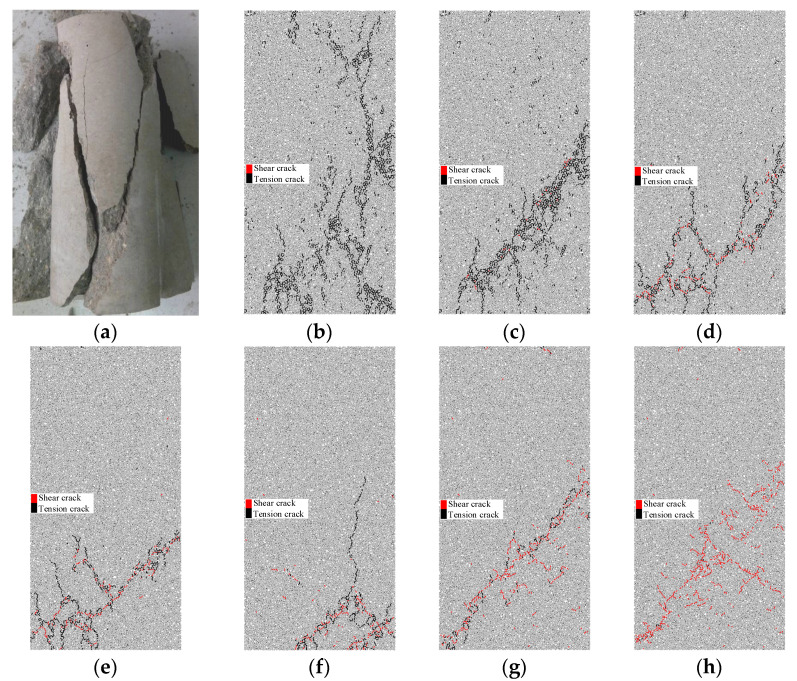
Comparison of discrete fractures under different values of the bond ratio: (**a**) compressive failure of sandstone; (**b**) ${\bar{r}}_{s}=0.1$; (**c**) ${\bar{r}}_{s}=0.5$; (**d**) ${\bar{r}}_{s}=1.0$; (**e**) ${\bar{r}}_{s}=1.2$; (**f**) ${\bar{r}}_{s}=1.5$; (**g**) ${\bar{r}}_{s}=2.0$; (**h**) ${\bar{r}}_{s}=5.0$.

**Figure 8 materials-15-05164-f008:**
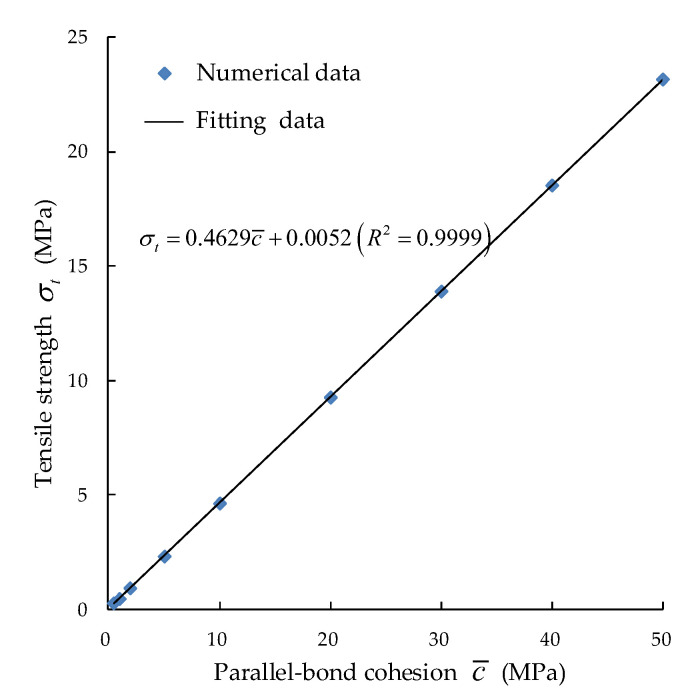
Corresponding relation between tensile strength and parallel bond cohesion.

**Figure 9 materials-15-05164-f009:**
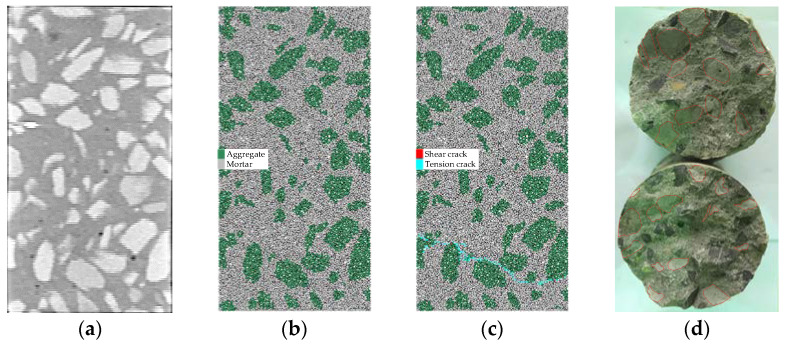
Concrete specimen: (**a**) CT scanning image of the specimen section; (**b**) PFC numerical model; (**c**) tension crack development; (**d**) fracture after tensile failure.

**Figure 10 materials-15-05164-f010:**
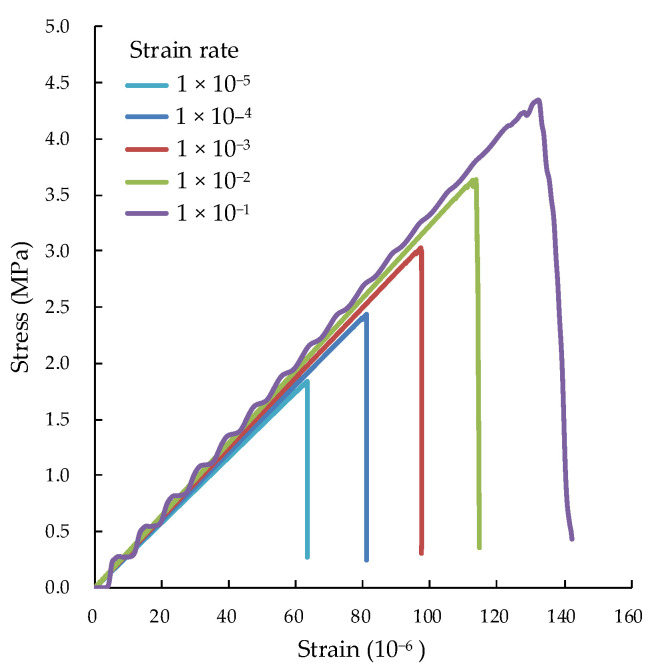
Numerical results of stress–strain curves of concretes under different strain rates.

**Figure 11 materials-15-05164-f011:**
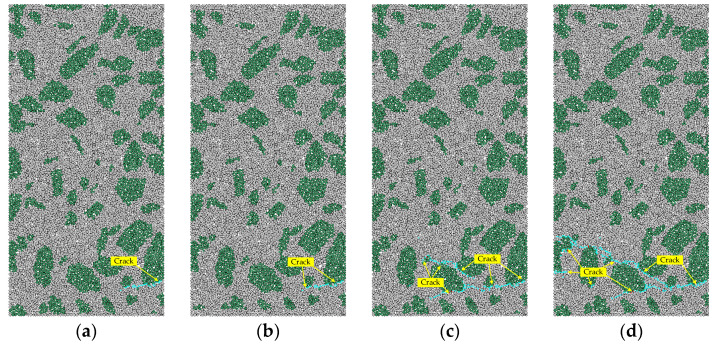
The crack propagation process of concrete under tension failure. (**a**) 95% peak stress; (**b**) 100% peak stress; (**c**) 70% peak stress after peak; (**d**) 10% peak stress after peak.

**Figure 12 materials-15-05164-f012:**
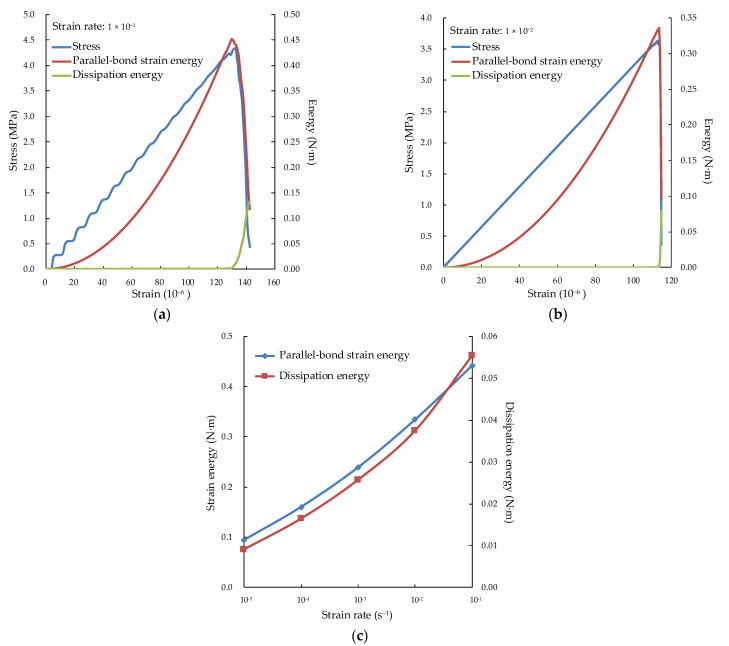
Energy evolution under different strain rates: (**a**) energy evolution at strain rate 1 × 10^−1^; (**b**) energy evolution at strain rate 1 × 10^−2^; (**c**) comparison of strain energy and dissipated energy.

**Figure 13 materials-15-05164-f013:**
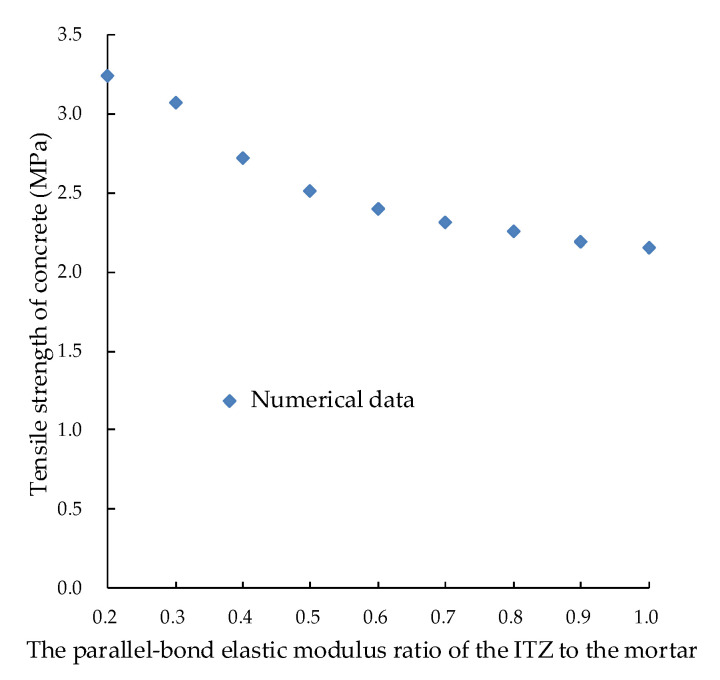
Relation between the parallel bond elastic modulus ratio of the ITZ to the mortar and the tensile strength of concrete.

**Table 1 materials-15-05164-t001:** Tensile test results of specimens under different strain rates.

Specimen	Number	Loading Rate (m/s)	Strain Rate (s^−1^)	Average Strain Rate (s^−1^)	Tensile Strength (MPa)	Average Tensile Strength (MPa)	Strain at Peak Strength (10^−6^)	Elastic Modulus (GPa)
Mortar	M4-1	1.00 × 10^−4^	9.49 × 10^−5^	8.60 × 10^−5^	4.45	3.19	189.1	24.44
	M4-2		6.60 × 10^−5^		2.03		103.8	21.44
	M4-3		9.72 × 10^−5^		3.08		150.4	23.53
	M5-1	1.00 × 10^−5^	8.62 × 10^−6^	9.06 × 10^−6^	2.13	2.84	101.6	21.9
	M5-2		1.00 × 10^−5^		3.82		180.2	22.34
	M5-3		8.56 × 10^−6^		2.56		123.7	23.02
	M6-1	1.00 × 10^−6^	8.92 × 10^−7^	8.46 × 10^−7^	2.09	2.19	100.9	23.53
	M6-2		8.54 × 10^−7^		2.35		113.2	22.99
	M6-3		7.93 × 10^−7^		2.14		111.3	21.78
Interface	I4-1	1.0 × 10^−4^	4.63 × 10^−5^	4.66 × 10^−5^	3.10	2.13	90.9	44.72
	I4-2		3.63 × 10^−5^		1.18		50.5	37.36
	I4-3		5.73 × 10^−5^		2.11		62.7	37.25
	I5-1	1.0 × 10^−5^	6.95 × 10^−6^	5.90 × 10^−6^	1.56	1.37	46.5	34.12
	I5-2		5.71 × 10^−6^		1.19		40.1	31.68
	I5-3		5.03 × 10^−6^		1.35		50.6	30.40
	I6-1	1.0 × 10^−6^	6.35 × 10^−7^	6.10 × 10^−7^	1.08	1.01	35.7	31.77
	I6-2		7.26 × 10^−7^		1.27		43.2	36.61
	I6-3		4.70 × 10^−7^		0.67		25.5	38.68
Concrete	C3-1	1.0 × 10^−3^	5.70 × 10^−4^	6.17 × 10^−4^	3.22	3.34	104.26	35.1
	C3-2		6.30 × 10^−4^		3.35		105.22	34.3
	C3-3		6.50 × 10^−4^		3.46		111.28	33.1
	C4-1	1.0 × 10^−4^	5.80 × 10^−5^	6.23 × 10^−5^	2.28	2.59	77.98	30.2
	C4-2		6.40 × 10^−5^		2.84		80.13	30.1
	C4-3		6.50 × 10^−5^		2.64		96.25	29.9
	C5-1	1.0 × 10^−5^	5.50 × 10^−6^	7.08 × 10^−6^	2.24	2.34	88.61	36.5
	C5-2		5.65 × 10^−6^		2.03		87.77	33.7
	C5-3		1.01 × 10^−5^		2.76		102.27	30.9

**Table 2 materials-15-05164-t002:** Macro- and meso-parameters of materials under different strain rates.

Strain Rate (s^−1^)	Macro Strength of Mortar (MPa)		Meso Cohesion of Interface (MPa)		Elastic Modulus of Mortar (GPa)	
	Macro	Meso	Macro	Meso	Macro	Meso
1 × 10^−5^	2.76	5.95	1.67	3.6	22.84	31.94
1 × 10^−4^	3.28	7.07	2.22	4.78	23.7	33.11
1 × 10^−3^	3.8	8.19	2.76	5.95	24.57	34.29
1 × 10^−2^	4.31	9.31	3.3	7.12	25.43	35.46
1 × 10^−1^	4.83	10.43	3.84	8.29	26.30	36.64

**Table 3 materials-15-05164-t003:** Relative error between the fitting value and the numerical value of concrete under different strain rates.

Strain Rate (s^−1^)	Tensile Strength (MPa)				Elastic Modulus (GPa)			
	Fitting Value	Numerical Value	Root Mean Square Error	Margin of Deviation	Fitting Value	Numerical Value	Root Mean Square Error	Margin of Deviation
1 × 10^−5^	1.95	1.84	0.11	0.19	29.33	29.45	0.12	0.29
1 × 10^−4^	2.41	2.44	0.03	0.24	30.47	30.53	0.05	0.30
1 × 10^−3^	2.88	3.03	0.15	0.28	31.62	31.47	0.15	0.31
1 × 10^−2^	3.34	3.64	0.30	0.33	32.76	32.67	0.10	0.32
1 × 10^−1^	3.81	4.35	0.54	0.38	33.91	33.66	0.25	0.33

## Data Availability

Not applicable.

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
