# Peer review of "Study on Axial Tensile Strain Rate Effect on Concrete Based on Experimental Investigation and Numerical Simulation"

_materials, 2022, doi:10.3390/ma15155164_

Round 1
Reviewer 1 Report
After going through the manuscript " Study on axial tensile strain rate effect on concrete based on experimental investigation and numerical simulation", I would give my comments below.
The authors reported an experimental investigation of the axial tension tests of mortar, interface, and concrete specimens under different strain rates according to the range and characteristics of strain rate under earthquake.
This paper has an interesting theme, but there are several critical problems:
- First of all, what’s the novelty of this work. Several works published in this category. What is different this work from other same papers?
- Structure of the introduction this paper is similar to the technical report, not an academic paper, so the authors should again rewrite some of the parts base on the journal paper style. The structure of the article needs more coherence.
- The parallel-bond elastic modulus ratio of the ITZ needs validation.
- The methodology section is not well organized for the readers to understand the concept.
- The quality of figures 9 and 11 CT are poor. The highlights of the work should be distinguished.
- what’s target of this part:
“Obviously, the ITZ reduces the tensile strength of the concrete. The tensile strength and elastic modulus of the concrete calculated by numerical simulation are 2.32 MPa and 30.0 GPa, respectively, which are in good agreement with the experimental values.”
- Conclusion is shallow, and authors are expected to describe more details.
- There are some grammatical errors, please carefully check the whole manuscript.
- The authors should report the values of mean square error and margin of deviation for curves.
e structure of the article needs more coherence.
- The methodology section is not well organized for the readers to understand the concept.
The presentation of the manuscript is lengthy and should be improved with concise information. The authors are requested to correct the abstract section with brief information without sub-paragraphs, for example, the aim of the review, how this review address various issues, and its applications in the field to the general audience. The literature flow of the review is satisfactory. Overall, it is a good study and therefore recommended for publication in Buildings after major revision.
Author Response
Dear Reviewer:
Thank you for your comments and suggestions. We have modified according to your comments and suggestions. The font color of the modified part has been marked red in the Revised Manuscript. the modification instructions have been uploaded.
Kind regards,
Nan Wu

Reviewer 2 Report
Quasi-static experiments were carried out in the work in the range of strain rates 0.000001 - 0.1 1/s. The authors suggest that the growth of deformation diagrams and tensile strength in the tested range of strain rates is a manifestation of dynamics. This is categorically false. If you look in table 3, the error in the value of the strength of the material, as the maximum value on the stress-strain curve, is large enough for all strain rates. If we carry out deformation dependences taking into account the error, then it turns out that all dependences are in the same range. Therefore, it is difficult to assert any dynamics. Considering that the idea of the alleged dynamics of the dynamic properties of concrete is key and incorrect from a scientific point of view, the presented article cannot be accepted. I recommend removing the idea of dynamics from the article, since the strain rates are quasi-static, not dynamic.
Author Response

(The authors gave the same response as above.)

Reviewer 3 Report
The following information should be added to the revised paper.
1. Section 2.1
The following experimental information should be added:
(1) The number of specimens;
(2) The concrete compressive strength of each specimen;
(3) The dimensional details of specimens.
2. Chapter 3.
Stress-strain curves for all specimens should be provided.
3. p.4, Table 1
In Table 1, the meanings of Specimen (Mortar, Interface, concrete) and Number are ambiguous. Please add a detailed description to the revised paper.
4. p.6, lines 162-165
Information on specimen details and test method for uniaxial compression test of mortar should be added.
5. Figure 7
(1) In Figure 7, it is necessary to explain how the shear crack and the tension crack are distinguished.
(2) A more detailed explanation of how to obtain Figure 7 is needed.
6. Section 5.1
The amount of error between the experimental and analytical results should be provided using a table or graph.
Author Response

(The authors gave the same response as above.)

Round 2
Reviewer 1 Report
The manuscript quality has been improved.
Reviewer 2 Report
Accept
Reviewer 3 Report
Accept in present form.
This manuscript is a resubmission of an earlier submission. The following is a list of the peer review reports and author responses from that submission.